# Predator-Prey Interactions between *Halobacteriovorax* and Pathogenic *Vibrio parahaemolyticus* Strains: Geographical Considerations and Influence of *Vibrio* Hemolysins

Gary P. Richards,[a] Michael A. Watson,[a] Henry N. Williams,[b] Jessica L. Jones[c]

[a]U.S. Department of Agriculture, Agricultural Research Service, Delaware State University, Dover, Delaware, USA
[b]School of the Environment, Florida Agricultural and Mechanical University, Tallahassee, Florida, USA
[c]U.S. Food and Drug Administration, Division of Seafood Science and Technology, Gulf Coast Seafood Laboratory, Dauphin Island, Alabama, USA

**ABSTRACT** *Halobacteriovorax* is a genus of naturally occurring marine predatory bacteria that attack, replicate within, and lyse vibrios and other bacteria. This study evaluated the specificity of four *Halobacteriovorax* strains against important sequence types (STs) of clinically relevant *Vibrio parahaemolyticus*, including pandemic strains ST3 and ST36. The *Halobacteriovorax* bacteria were previously isolated from seawater from the Mid-Atlantic, Gulf of Mexico, and Hawaiian coasts of the United States. Specificity screening was performed using a double agar plaque assay technique on 23 well-characterized and genomically sequenced *V. parahaemolyticus* strains isolated from infected individuals from widely varying geographic locations within the United States. With few exceptions, results showed that *Halobacteriovorax* bacteria were excellent predators of the *V. parahaemolyticus* strains regardless of the origins of the predator or prey. Sequence types and serotypes of *V. parahaemolyticus* did not influence host specificity, nor did the presence or absence of genes for the thermostable direct hemolysin (TDH) or the TDH-related hemolysin, although faint (cloudy) plaques were present when one or both hemolysins were absent in three of the *Vibrio* strains. Plaque sizes varied depending on both the *Halobacteriovorax* and *Vibrio* strains evaluated, suggesting differences in *Halobacteriovorax* replication and/or growth rates. The very broad infectivity of *Halobacteriovorax* toward pathogenic strains of *V. parahaemolyticus* makes *Halobacteriovorax* a strong candidate for use in commercial processing applications to enhance the safety of seafoods.

**IMPORTANCE** *Vibrio parahaemolyticus* is a formidable obstacle to seafood safety. Strains pathogenic to humans are numerous and difficult to control, especially within molluscan shellfish. The pandemic spread of ST3 and ST36 has caused considerable concern, but many other STs are also problematic. The present study demonstrates broad predatory activity of *Halobacteriovorax* strains obtained along U.S. coastal waters from the Mid-Atlantic, Gulf Coast, and Hawaii toward strains of pathogenic *V. parahaemolyticus*. This broad activity against clinically relevant *V. parahaemolyticus* strains suggests a role for *Halobacteriovorax* in mediating pathogenic *V. parahaemolyticus* levels in seafoods and their environment as well as the potential application of these predators in the development of new disinfection technologies to reduce pathogenic vibrios in molluscan shellfish and other seafoods.

**KEYWORDS** host specificity, *Halobacteriovorax*, *Vibrio parahaemolyticus*, predator, hemolysin, geographical, prey, sequence type

Address correspondence to Gary P. Richards, gary.richards@usda.gov.

The authors declare no conflict of interest.

*[This article was published on 6 July 2023 with an error in the supplemental material. The supplemental material was corrected in the current version, posted on 17 August 2023.]*

*B*dellovibrio and like organisms (BALOs) are a group of small, polar-flagellated, obligate predatory bacteria that prey upon and kill a variety of principally Gram-negative bacteria. They are currently classified under four genera (*Bdellovibrio*, *Bacteriovorax*, *Peredibacter*, and *Halobacteriovorax*) (1). *Halobacteriovorax* contains the only marine species with widely varying

salt requirements and tolerances (reviewed in reference 1). The life cycle of *Halobacteriovorax* involves an attack-phase predator that seeks out, attacks, and digests a hole in the cell wall of susceptible prey bacteria through which it enters the intraperiplasmic space, forming a predator-prey structure called a bdelloplast. The prey dies, and the *Halobacteriovorax*, using host nutrients, elongates until the nutrients are depleted, at which time it septates and divides into progeny cells. The progeny are released into the extracellular milieu after lysis of the former host's cell wall to initiate another cycle of infection (2, 3).

Previous specificity studies indicate that *Vibrio parahaemolyticus* is a common prey for *Halobacteriovorax* (1, 4, 5). This, in conjunction with the ubiquitous distribution of *Halobacteriovorax* in salt water (1, 6), suggests that *Halobacteriovorax* may be one of nature's tools to limit the proliferation of *V. parahaemolyticus* populations in the marine environment (1, 7). However, many of the susceptibility studies on the predation of *V. parahaemolyticus* by *Halobacteriovorax* have used environmental *Vibrio* isolates, not clinically isolated *V. parahaemolyticus* strains. There are many strains of *V. parahaemolyticus* that cause shellfish-associated illnesses, particularly from the consumption of raw or undercooked oysters, as well as from other raw seafoods. Unfortunately, there is only limited information available on the susceptibility of specific sequence types (STs) and serotypes of human-pathogenic *V. parahaemolyticus* to predation by *Halobacteriovorax*.

*Vibrio parahaemolyticus* is widespread within the marine environment throughout temperate and tropical regions of the world. It is considered a common cause of seafood-related bacterial illnesses. An estimate of the number of reported and underreported foodborne cases of *V. parahaemolyticus* acquired in the United States annually ranges between 34,664 and 58,027 (8). In a study involving clinical isolates of *V. parahaemolyticus* from North America, Jones et al. (9) identified numerous strains from stools and wounds of infected individuals, as well as apparently nonclinical isolates derived directly from oysters. They also determined whether the strains contained potential virulence factors, including the thermostable direct hemolysin gene (*tdh*) and the thermostable direct hemolysin-related hemolysin gene (*trh*) (10). Those results were provided in more detail for these same strains in a recent paper by Miller et al. (11), who also reported genomic sequences for all the *V. parahaemolyticus* isolates.

This study evaluates four strains of *Halobacteriovorax* obtained from the Gulf of Mexico, Hawaii, and tributaries of the Delaware Bay (12, 13) to determine their ability to prey upon and kill 23 well-characterized and clinically relevant *V. parahaemolyticus* isolates from the Jones et al. study (9). The vibrios represent six STs, including multiple ST3 and ST36 strains obtained from around the United States. The ST3 and ST36 strains are pandemic strains that include serotypes O3:K6 and O4:K12, respectively (14–18). Two ST631 strains are also included in the evaluation. ST631 is known to be endemic in northern states (19, 20). We also evaluated whether *Halobacteriovorax* strains preferentially infect *V. parahaemolyticus* strains that are common to the same geographical area or if they are broadly infectious toward *V. parahaemolyticus* isolates from other, more distant habitats. Additionally, we evaluated the infectivity of each *Halobacteriovorax* isolate toward *V. parahaemolyticus* strains in the presence and absence of genes for virulence-associated hemolysins to determine if the hemolysins might protect the vibrios from predation.

## RESULTS AND DISCUSSION

**_Halobacteriovorax_ isolates and BLAST searches.** Four *Halobacteriovorax* isolates were originally obtained from seawater from sites along the Delaware Bay (Mid-Atlantic), the Gulf Coast of Alabama, and Hawaii (13). Strain designations, state from which each isolate was obtained, and coordinates are as follows: G3, Alabama, 30°15′25.57″N, 88°6′21.73″W; OS1, Delaware, 38°47′26.37″N, 75°09′51.36″W; S11, Delaware, 39°05′05.94″N, 75°27′39.99″W; and H4, Hawaii, 19°43′42.9″N, 156°3′46.2″W. PCR of a portion of the 16S rRNA gene was performed on each isolate using new primers designed as part of this study. Primers HBx3aF (5′-GAAACCCTGACGCAGCAATG-3′) and HBx4bR (5′-TTTCGCGCCTCAGCGTCAGTT-3′) produced a single band of ~390 bp. These primers were designed to avoid amplification of *V. parahaemolyticus* DNA that may have carried over from the *Halobacteriovorax* enrichments.

**TABLE 1** Percent identity of partial 16S rRNA gene sequences among four *Halobacteriovorax* strains as determined by two-way BLAST searches

| Strain | Homology (%) | | | |
| --- | --- | --- | --- | --- |
| | OS1 | H4 | G3 | S11 |
| S11 | 99.69 | 99.08 | 89.63 | 100.00 |
| G3 | 89.30 | 89.60 | 100.00 | |
| H4 | 99.38 | 100.00 | | |
| OS1 | 100.00 | | | |

These primers were also designed to bind to complementary regions in the three named species of *Halobacteriovorax* found in GenBank (*H. marinus*, *H. litoralis*, and *H. vibrionivorans*). PCR controls against the possible amplification of carryover DNA from the *V. parahaemolyticus* enrichments were consistently negative. Sequencing confirmed the isolates were different from each other in side-by-side BLAST comparisons (Table 1). Partial 16S rRNA sequences for the four *Halobacteriovorax* isolates are given in Table S1 in the supplemental material. These sequences were also run on MegaBLAST against the *Halobacteriovorax* (taxid identifier [ID] 1652133) database. Results are shown in Table 2. Three of the isolates were most closely aligned with *Halobacteriovorax* sp. strain PA1, while one of the species (G3) showed 91.69% identity toward three *Halobacteriovorax* strains (*H. vibrionivorans*, *Halobacteriovorax* sp. strain Y22, and *Halobacteriovorax* sp. strain BALOs_7). Highly similar 16S rRNA sequences of *Halobacteriovorax* strains and the limited number of named species in GenBank prevent definitive identification of the isolates.

**Vibrio parahaemolyticus strains.** Twenty-three clinical strains of *V. parahaemolyticus*, as previously described by Jones et al. (9) and Miller et al. (11), were used as potential prey for the four *Halobacteriovorax* strains. The vibrios were kindly provided by the U.S. Food and Drug Administration (FDA). Genomic sequences of each of these isolates were previously determined and are given in the work of Miller et al. (11). Sequence types and serotypes of these strains and the state where each illness was reported are shown in Fig. 1.

**Specificity studies and geographic implications.** The ability of *Halobacteriovorax* to predate upon different clinical strains of *V. parahaemolyticus* obtained from different geographic areas was determined by plaque assay. Results show that most *Halobacteriovorax* strains are predatory toward the vibrios (Fig. 1). Very broad specificity was observed for the *Halobacteriovorax* strains; all four strains predated upon 21 or 22 of the 23 *V. parahaemolyticus* strains plus the *V. parahaemolyticus* strain RIMD 2210633 (ST3, O3:K6 pandemic strain that was isolated in Japan) (Fig. 1). The RIMD strain was used for our original isolation of the *Halobacteriovorax* strains from seawater (5). Only two of the *V. parahaemolyticus* strains (CDC strain K5582, an ST631 isolate from Georgia, and K4859, an untypeable strain from Hawaii) were resistant to predation by three of the four *Halobacteriovorax* isolates (Fig. 1). The K5582 isolate from Georgia was infected only by *Halobacteriovorax* G3, which was isolated from seawater from the neighboring state of Alabama, while the K4859 strain from Hawaii was infected only by the Hawaiian strain of *Halobacteriovorax* (H4). Three other *Vibrio* strains from Georgia and another three from Hawaii did not show any predator-prey preferences based on the locations from which the *Halobacteriovorax* and vibrios were isolated (Fig. 1).

Results showed generally broad specificity of *Halobacteriovorax* isolates from different regions of the United States against strains of *V. parahaemolyticus* from widely varying

**TABLE 2** Top hits and percent identities of four *Halobacteriovorax* strains sequenced in this study as determined by MegaBLAST searches of partial 16S rRNA gene sequences

| Strain | Homology (%)[a] | Top hit | GenBank accession no. for top hit |
| --- | --- | --- | --- |
| S11 | 99.08 | *Halobacteriovorax* sp. PA1 | KR493097.1 |
| G3 | 91.69 | *H. vibrionivorans* | MH590697.1 |
| G3 | 91.69 | *Halobacteriovorax* sp. BALOs_7 | CP027772.1 |
| G3 | 91.69 | *Halobacteriovorax* sp. Y22 | MH997664.1 |
| H4 | 100.00 | *Halobacteriovorax* sp. PA1 | KR493097.1 |
| OS1 | 99.38 | *Halobacteriovorax* sp. PA1 | KR493097.1 |

[a]MegaBLAST search was performed against the *Halobacteriovorax* database (taxid ID 1652133).

| *Vibrio parahaemolyticus* strain information | | | | Plaques produced on *V. parahaemolyticus* by four *Halobacteriovorax* strains, plaque sizes, and plaque clarity[a] | | | | Hemolysin gene presence[b] | |
|---|---|---|---|---|---|---|---|---|---|
| CDC Strain | Sequence type | Serotype | Strain origin | OS1 | S11 | G3 | H4 | *tdh* | *trh* |
| K4637-1 | 3 | O3:K6 | NY | + S | + P S | + S | + S | + | + |
| K4775 | 3 | O3:K6 | GA | + S | + P | + S | + P | + | - |
| K5058 | 3 | O3:K6 | TX | + P S | + P S | + S | + S M | + | - |
| K5010-1 | 3 | O1:Kuk[c] | MA | + S | + S | + S M | + S | + | - |
| K5528 | 3 | O4:K68 | GA | + S    F | + S    F | + S | + S | + | - |
| K4639-1 | 36 | O4:K12 | NY | + S M | + P S | + S M | + S | + | + |
| K5278 | 36 | O4:K12 | WA | + S M | + S | + M | + M L | + | + |
| K5281 | 36 | O4:K12 | WA | + S M | + S | + M L | + M L | + | + |
| K5308 | 36 | O4:K63 | AK | + M | + S | + S M | + S M | + | + |
| K5345-1 | 36 | O4:K12 | IA | + M | + S | + S M | + S M | + | + |
| K5346 | 36 | O4:K12 | PA | + M | + S | + S M | + S M | + | + |
| K5433 | 36 | O4:Kuk | WA | + M | + S | + M | + M | + | + |
| K5437 | 36 | O4:Kuk | WA | + M | + S | + S M | + S M | + | + |
| K5512 | 36 | O4:K12 | OK | + M | + S | + S M | + S M | + | + |
| K5629 | 36 | O4:K13 | GA | + M | + S | + S M | + S M | + | + |
| K5638 | 36 | O4:K12 | MD | + S | + S | + L | + S | + | + |
| K5435 | 65 | O1:Kuk | WA | + S    F | + S    F | + L | + S M    F | - | + |
| K4857-1 | 79 | O5:K17 | HI | + M | + S | + S M | + S M | - | - |
| K4858 | 283 | O4:K4 | HI | + S M | + S | + M | + S M | - | - |
| K5276 | 631 | O11:Kuk | NY | + M L | + S | + L | + M | + | + |
| K5582 | 631 | O11:Kuk | GA | - | - | + L | - | + | + |
| K4859 | Unt[c] | O4:K18 | HI | - | - | - | + P S | - | + |
| K5282 | Unt | O5:Kuk | HI | + M L    F | + S    F | + L | + M L    F | - | - |
| Host Vp RIMD 2210633[d] | 3 | O3:K6 | Japan | + M L | + S M | + L | + L | + | - |

**FIG 1** Specificity of four *Halobacteriovorax* isolates toward clinical strains of *Vibrio parahaemolyticus*, plus relative plaque sizes, plaque clarity, and influence of *Vibrio* hemolysins. [a]The ability of *Halobacteriovorax* strains to infect strains of *V. parahaemolyticus* is indicated by + or −. The locations from which *Halobacteriovorax* strains were obtained are as follows: OS1 and S11, Delaware Bay; G3, Gulf of Mexico on the coast of Alabama; and H4, Hawaii. Relative plaque sizes (represented by lettered and colored blocks) are as follows: pinpoint (P, gray box), small (S, red box), medium (M, blue box), and large (L, yellow box). If populations of two distinctly different-sized plaques were observed, then both sizes are listed by two colored and lettered blocks. Plaques were clear unless marked with an "F" (green box), which signifies faint (cloudy) plaques. [b]Presence or absence of the *tdh* gene for the thermostable direct hemolysin (TDH) and the *trh* gene for the TDH-related hemolysin as determined by PCR and reported by Jones et al. (9) and Miller et al. (11). [c]Abbreviations: "Unt" indicates that the sequence type is untypeable using the typing method of González-Escalona et al. (21); "Kuk" signifies that the K serotype is untypeable. [d]Host Vp is the *V. parahaemolyticus* strain that was used to originally isolate the four *Halobacteriovorax* strains used in this study.

locations within the United States. This is clearly observed for *Vibrio* isolates from the West Coast (four ST36 and one ST65 isolate, all from Washington State, and an ST36 isolate from Alaska), which were infected by East Coast, Gulf Coast, and Hawaiian strains of *Halobacteriovorax* (Fig. 1). Predation of *V. parahaemolyticus* from Washington State and Alaska by *Halobacteriovorax* from widely differing geographic areas is further evidence that local habitats are not essential for most predator-prey interactions. Although the origin within the United States did not generally restrict the ability of *Halobacteriovorax* to invade *V. parahaemolyticus* from other regions of the United States, it is unclear whether *Halobacteriovorax* from the United States would prey upon strains of *V. parahaemolyticus* from more distant locations. It is apparent, however, that all four U.S. strains of *Halobacteriovorax* readily infected *V. parahaemolyticus* RIMD 2210633, the host strain that was originally isolated in Japan.

Since *Halobacteriovorax* bacteria are not known to replicate outside a host bacterium, and the above-described assays used the original host bacterium (*V. parahaemolyticus* RIMD 2210633) to initially isolate and subsequently propagate the predators, it was necessary to remove the host cells before performing the specificity testing in other *V. parahaemolyticus* strains. This was accomplished using an enrichment, filtration, dilution method as previously published (5). The resulting *Halobacteriovorax* stocks were tested to ensure successful removal of the vibrios in the enrichment culture. Controls showed no *Vibrio* contamination in any of the filtered and diluted *Halobacteriovorax* stocks used in our host specificity studies.

**Prey specificity as affected by sequence type and serotype.** Predator-prey interactions were determined with six different sequence types and 12 serotypes of *V. parahaemolyticus*. Results for five isolates of ST3 and 11 isolates of ST36 demonstrate predator-prey interactions (Fig. 1). The ST3 isolates comprised serotypes O3:K6 (3 each), O4:K68 (1 each), and O1:Kuk, where Kuk stands for K untypeable (1 each). *Halobacteriovorax* preyed upon the five ST3 strains and an additional ST3 strain used as the original host (*V. parahaemolyticus* RIMD 2210633). The ST36 isolates consisted of serotypes O4:K12 (7 each), O4:K13 (1 each), O4:Kuk (2 each), and O4:K63 (1 each). Neither the ST nor the serotype appeared to affect the predation of *Halobacteriovorax* on any of these *V. parahaemolyticus* strains. The *Halobacteriovorax* bacteria were also quite capable of infecting other STs and serotypes of clinically important *V. parahaemolyticus* isolates (Fig. 1).

**Plaque sizes.** Specific plaque diameters were not measured here because sizes varied depending somewhat on the density of the plaques. Many plaque assay plates contained plaques that were too numerous to count, where most of the plaques could not be individually discriminated. Plaques were first visible usually 2 to 3 days postplating, or slightly longer for faint plaques, and slowly increased in size over the 5- to 7-day incubation period. Where individual sizes were observable, plaques ranged from pinpoint (~1 mm in diameter) to large (~1 cm in diameter). A breakdown of relative plaque sizes (pinpoint, small, medium, and large) is shown by the colored and lettered blocks in Fig. 1. Relative plaque sizes give some clues about the growth and replication rates of individual *Halobacteriovorax* strains on any given *V. parahaemolyticus* host strain under the prevailing conditions. As evident from Fig. 1, *Halobacteriovorax* G3 produced small, medium, and large plaques depending on the *Vibrio* strain. In contrast, *Halobacteriovorax* S11 produced only pinpoint or small plaques regardless of the *Vibrio* strain, except in the original host (RIMD 2210633), where plaque sizes were small to medium. Most of the pinpoint-sized plaques were found in the ST3, O3:K6 vibrios, while *Halobacteriovorax* strains OS1, G3, and H4 produced small and/or medium-sized plaques in the majority of ST36 strains, with occasional large plaques as well (Fig. 1). Based on the larger plaques on the ST36 strains than on the ST3 strains, it appears that *Halobacteriovorax* more efficiently attacked or replicated in ST36 than in ST3 strains. This suggests that some *Vibrio* sequence types may contain resistance factors against certain predators.

Large plaques were most often produced by *Halobacteriovorax* G3 and were about twice as likely to form with G3 as with OS1 or H4. Large plaques were also produced by OS1, G3, and H4 in the original host *V. parahaemolyticus* RIMD 2210633. No large or medium plaques were produced by S11 in the *Vibrio* strains, except in the original host *Vibrio*, which had small and medium plaques. Some predators, like S11, may invade and/or reproduce within the vibrios more slowly, perhaps due to suboptimal growth conditions for a particular isolate. Previously, we showed that the salinity of the plaque assay medium can affect plaque sizes, with smaller plaques formed when the salinity of the medium is higher than the salinity of the seawater from which the *Halobacteriovorax* bacteria were originally obtained (5). In the present study, plaque assay medium contained approximately 3% NaCl (30 ppt) and the salinities of the seawater from which the S11 and OS1 strains were originally obtained were 23.1 ppt and 25.4 ppt, respectively. This 2.3-ppt difference in salinity may be the reason the OS1 plaques were occasionally larger than the S11 plaques. Both strains came from neighboring locations along the Delaware Bay (12).

**Clear versus faint (cloudy) plaques.** Plaques were clear unless designated faint (or cloudy) as indicated in Fig. 1 by green boxes labeled "F." The cloudy plaques likely represent incomplete killing of the host vibrios, perhaps due to the presence or development of *Vibrio*

resistance factors by some of the *V. parahaemolyticus* strains. Faint plaques remained cloudy from the first visualization of the plaques (usually 3 to 4 days) until the end of the incubation period (7 days). The faint plaques were associated with three *V. parahaemolyticus* strains (K5528, K5435, and K5282) that were isolated from Georgia, Washington State, and Hawaii, respectively, following challenge with *Halobacteriovorax* isolates obtained from the Mid-Atlantic and Hawaii (OS1, S11, and H4). The Gulf strain (G3) did not produce faint plaques in any of the *Vibrio* strains. Faint plaques were present in one of the five strains of ST3 but none of the 11 strains of ST36. Faint plaques were present only in the absence of one or both *V. parahaemolyticus* hemolysins. More information about the effect of the lack of specific hemolysins on *Halobacteriovorax* infectivity is discussed below.

**Influence of *Vibrio* hemolysins on *Halobacteriovorax* predation.** *Vibrio parahaemolyticus* contains hemolysins that are potential virulence factors in humans. They include the thermostable direct hemolysin (TDH) and the TDH-related hemolysin (TRH), encoded by the *tdh* and *trh* genes, respectively. Results of host specificity studies were compared with the presence or absence of *tdh* and/or *trh* (Fig. 1). *Halobacteriovorax* successfully attacked most of the vibrios regardless of hemolysin presence or absence. Fourteen of the *Vibrio* strains were both *tdh* and *trh* positive, of which 13 strains were readily attacked by all four predators. The exception was CDC strain K5582, an ST631 strain, which was susceptible to only *Halobacteriovorax* G3. Three of the *V. parahaemolyticus* strains were both *tdh* and *trh* negative and were predated upon by all *Halobacteriovorax* strains (Fig. 1). Only five of the *V. parahaemolyticus* strains were *tdh* negative, four of which were strains isolated in Hawaii. Only one of the *tdh*-negative strains (CDC K4859 from Hawaii) was resilient to predation by three of the four predators. The other four vibrios were susceptible to attack by all four *Halobacteriovorax* strains. Five of the vibrios were *tdh* positive and *trh* negative, including the host *Vibrio* RIMD 2210633, and were all predated upon by the four *Halobacteriovorax* strains. Four of these were ST3, which is commonly observed to be *tdh* positive and *trh* negative (21, 22). Conversely, only two of the vibrios were *tdh* negative and *trh* positive, of which one was attacked by all four predators but the other was attacked only by H4. Overall, it appears that these two hemolysin genes, either separately or together, do not prevent plaque formation on *V. parahaemolyticus*. However, faint plaques were produced only when either or both the *tdh* and/or *trh* genes were absent, specifically for *Vibrio* strains K5528 (*tdh* positive/*trh* negative), K5435 (*tdh* negative/*trh* positive), and K5282 (*tdh* negative/*trh* negative) from Georgia, Washington State, and Hawaii, respectively (Fig. 1). The same results were consistently obtained upon repeated assays. Therefore, it appears that the absence of these *Vibrio* hemolysins contributes to survival of a portion of the vibrios, the likely source of the cloudiness in three of the *Vibrio* strains. For other *Vibrio* hosts, the presence or absence of one or both hemolysins appears to have no effects on *Halobacteriovorax* infectivity, since plaques were clear, indicating total or nearly total elimination of the vibrios. Further study of possible resistance factors in some strains of *V. parahaemolyticus* and virulence factors in *Halobacteriovorax* strains is warranted.

**Considerations of host specificity by *Halobacteriovorax* isolates.** Previously, we assessed the predatory ability of *Halobacteriovorax* strains OS1, OR7, S11, and G3 to prey upon five strains of *V. parahaemolyticus*, two strains of *Vibrio vulnificus*, and a single strain of *Vibrio alginolyticus* (5). The *Halobacteriovorax* strains were predatory toward all five *V. parahaemolyticus* strains tested at that time but were not predatory toward the other *Vibrio* species, thus demonstrating that these *Halobacteriovorax* strains have a strong affinity for *V. parahaemolyticus*. That was likely due, in part, to the fact that our *Halobacteriovorax* strains were originally isolated on *V. parahaemolyticus* strain RIMD 2210633. If we had originally isolated *Halobacteriovorax* strains specific for some other genus or species of bacterium, then they may not have shown an affinity toward *V. parahaemolyticus* strains, or their affinities may have been significantly reduced. Thus, the initial organism upon which *Halobacteriovorax* strains are isolated will likely bias the results toward selection of genus- or species-specific *Halobacteriovorax* strains. Clearly, other bacterial strains can serve as prey to *Halobacteriovorax* (reviewed in reference 1).

**_Halobacteriovorax_ bacteria as potential biocontrol agents.** The broad specificity of these *Halobacteriovorax* isolates toward *V. parahaemolyticus* from diverse and geographically

distinct regions of the United States is welcome news for the development of biocontrol measures against pathogenic strains of *V. parahaemolyticus*. These findings justify our further efforts to formulate a cocktail (mixture) of *Halobacteriovorax* strains for treating *Vibrio*-contaminated shellfish and other seafoods with the goals to reduce *V. parahaemolyticus* contamination and render the products safer to consume. Such treatments could have broad applications in reducing *V. parahaemolyticus* in market oysters, clams, and mussels using depuration, a commercial process employed internationally to reduce potential pathogens in contaminated shellfish (reviewed in reference 23). A simple pretreatment of the shellfish prior to depuration could be effective in reducing *V. parahaemolyticus* levels. Likewise, a simple dip of fish products in a solution containing *Halobacteriovorax* might safely kill vibrios on the surface of fish fillets without the need for either disinfecting chemicals or heat treatment. Based on our testing here, a cocktail containing *Halobacteriovorax* strains G3 and H4 should effectively target all the *V. parahaemolyticus* strains used in this study. Previous pilot studies have successfully demonstrated the efficacy of predatory bacteria in reducing bacterial pathogens in experimental infections in laboratory rats (24), plants (25), and seafoods, including fish (26, 27) and shellfish (28). Further research on the use of *Halobacteriovorax* and other predatory bacteria as potential biocontrol agents in food processing, disease prevention and treatment, and farming and aquaculture operations is warranted.

**Summary.** The specificities of four strains of *Halobacteriovorax* from the Mid-Atlantic, Gulf Coast, and Hawaii were determined on 23 clinical strains of *V. parahaemolyticus* obtained from regions around the United States. Broad predatory activity was observed with 21 or 22 of the 23 strains serving as prey. The remaining two vibrios appeared resistant to three of the four *Halobacteriovorax* strains, and reasons for this resistance are uncertain. *Halobacteriovorax* species obtained from three distinctly different geographic locations were readily capable of preying upon *V. parahaemolyticus* isolated from both similar and remote habitats. All five ST3 and 11 ST36 vibrio strains were susceptible to attack by the four *Halobacteriovorax* strains, providing some hope that these pandemic strains may be moderated, at least partially, by naturally occurring *Halobacteriovorax* in the marine environment. Plaques were generally larger in the ST36 than the ST3 strains, suggesting more rapid predator entry into its prey and/or faster replication and growth within ST36 strains. The Gulf Coast strain of *Halobacteriovorax* (G3) commonly produced the largest plaques. This is good news, since *V. parahaemolyticus* levels nationally are highest in the warmer waters of the Gulf, where the more rapidly replicating G3 strain is present. Finally, we saw no evidence that the presence or absence of the well-known *V. parahaemolyticus* hemolysin genes *tdh* and *trh* has any effects on *Halobacteriovorax*-induced plaque formation, although the lack of one or both genes may have led to the formation of faint plaques, suggesting only partial death of three of the *Vibrio* strains. Although the above-described studies cover only a thin slice of the many tasks needed to define *Halobacteriovorax* interactions with *V. parahaemolyticus* and other pathogenic species, our work provides strong evidence that pathogenic strains of *V. parahaemolyticus* are likely to be readily susceptible to *Halobacteriovorax* predation within the marine environment.

## MATERIALS AND METHODS

**Sources of bacterial isolates.** *Halobacteriovorax* strains were obtained from seawater from the coast of Alabama in the Gulf of Mexico (strain G3), riverine sites along the U.S. Mid-Atlantic coast of the Delaware Bay (strains OS1 and S11), and Keyhole Point near Kailua-Kona, HI (strain H4) (12, 13). They were isolated on *V. parahaemolyticus* strain RIMD 2210633, which was previously sequenced by Makino et al. (29) and for which chromosome sequences were entered into GenBank under accession numbers BA000031 and BA000032. This *Vibrio* is a pandemic strain isolated from Japan in 1996 and previously shown to be an ST3, serotype O3:K6 strain that is *tdh* positive and *trh* negative (29). Twenty-three clinical strains of *V. parahaemolyticus* were kindly provided by the U.S. Food and Drug Administration (FDA) under a material transfer agreement (MTA). These strains were previously described by Jones et al. (9) and Miller et al. (11) and are listed in Fig. 1. The isolates caused illnesses among individuals in a variety of U.S. states (Fig. 1). The *V. parahaemolyticus* and *Halobacteriovorax* isolates were maintained at −80°C as 30% glycerol stocks.

**Halobacteriovorax enrichments.** The *Halobacteriovorax* strains were enriched in their original host *V. parahaemolyticus* RIMD 2210633 in previously autoclaved and 0.22-$\mu$m-filtered, 30-ppt natural seawater for 24 to 48 h. After incubation, each enrichment was filtered through a 0.45-$\mu$m Acrodisc syringe filter (Pall Corp., Timonium, MD) and then diluted 1:1,000 in sterile 30-ppt natural seawater to dilute out any remaining bacterial host cells to extinction in accordance with our previously published enrichment, filtration, dilution

technique (5). Although filtration is usually sufficient to separate non-*Vibrio* host cells from the smaller *Halobacteriovorax*, filtration alone was shown to be ineffective in removing *Vibrio* minicells, which can pass through 0.45-$\mu$m filters (5). The 1:1,000 dilution of *Halobacteriovorax* was tested for residual vibrios by pour plate technique by plating 100 $\mu$L of each enrichment in LB agar containing 3% NaCl. This served as a control to ensure the *Halobacteriovorax* strains were not contaminated with their original host *Vibrio* prior to host specificity testing of the predator against new strains of *V. parahaemolyticus*. Any carryover of the original host *Vibrio* to new *Vibrio* strains has the potential to give false-positive results in host specificity assays.

**PCR and 16S rRNA gene sequencing of *Halobacteriovorax* isolates.** To ensure the four predators selected for analysis were all different strains of *Halobacteriovorax*, PCR primers were designed to amplify different *Halobacteriovorax* strains, but not residual host *Vibrio* DNA that could be present in the filtered enrichments, either from *Vibrio* minicells or from DNA from lysed vibrios, or both. Using the three named species of *Halobacteriovorax* in GenBank, (*H. marinus* [accession no. NR_102485.1], *H. litoralis* [NR_028724.1], and *H. vibrionivorans* [MH150810.1]), BLAST searches identified sequences that differed from the 16S rRNA gene of the host *V. parahaemolyticus* RIMD 2210633. Primers were synthesized by Integrated DNA Technologies, Coralville, IA. PCRs were performed on a Smart Cycler (Cepheid, Sunnyvale, CA) in 25-$\mu$L reaction mixtures containing SYBR green reverse transcription-quantitative PCR (RT-qPCR) kit reagents as recommended by the manufacturer (Sigma-Aldrich, St. Louis, MO), 200 nM (each) primer, and 1 $\mu$L of filtered *Halobacteriovorax* enrichment. Cycling conditions were as follows: *Taq* polymerase activation for 3 min at 95°C and 35 cycles at 94°C for 1 min, 51°C for 1 min, and 72°C for 1 min. Melt curves were obtained from 60 to 95°C with temperature increases of 0.2°C/s. Each amplicon was electrophoretically purified on a 1.5% high-gelling-temperature agarose (Fisher Scientific) gel, and fluorescent bands were excised and purified using a QIAquick gel extraction kit (Qiagen, Hilden, Germany) according to the manufacturer's instructions. DNA was submitted to Azenta US, Inc. (North Plainfield, NJ), for Sanger sequencing. The *Halobacteriovorax* sequences were compared in BLAST to determine the uniqueness of each strain as well as their similarities to other strains in GenBank.

**Predator-prey specificity assays.** In preparation for specificity assays using a double agar plaque assay procedure, the vibrios were subcultured by inoculation onto Difco thiosulfate citrate bile salts sucrose agar (TCBS; Becton, Dickinson and Company, Sparks, MD) plates and incubated overnight at 26°C. A colony was then picked and enriched in 10 mL of Difco Luria-Bertani (LB) broth (Becton, Dickinson and Company) containing 3% NaCl. Cultures were incubated at 37°C at 200 rpm to an optical density at 600 nm (OD$_{600}$) of $\sim$0.2. While the vibrios were incubating, bottom and top agars were prepared for plaque assays using autoclaved and 0.22-$\mu$m-filtered, 30-ppt natural seawater from the Delaware Bay. Bottom agar consisted of Bacto agar (15 g/L) and 0.1% polypeptone peptone medium (Pp20), both from Becton, Dickinson and Company, while the top agar consisted of Bacto agar (7.5 g/L) and 0.1% Pp20. After autoclaving, both agars were cooled to 50°C in a water bath. To prepare bottom agar plates, 20 mL was pipetted into sterile petri dishes. For top agar, 7.5 mL of autoclaved top agar was pipetted into sterile tubes. All tubes were maintained in the water bath at 50°C. After the bottom agar solidified, specificity testing on the *Vibrio* cultures was initiated when they reached an OD$_{600}$ of $\sim$0.2. At this time, 6.5 mL of sterile seawater, 100 $\mu$L of the filtered and 1:1,000-diluted *Halobacteriovorax* enrichments, and 1 mL of the *V. parahaemolyticus* culture were combined with the 7.5 mL of top agar, mixed by gentle inversion to prevent bubble formation, and poured over the bottom agar layer. After solidification, plates were inverted, incubated at 26°C, and monitored daily for the formation of plaques for 5 to 7 days, after which plaque presence or absence and appearance were recorded along with relative plaque sizes. Plates that contained pinpoint, small, or faint plaques were incubated for the full 7 days.

**Specificity comparisons.** Comparisons were drawn to evaluate (i) the ability of predators to invade and kill different *V. parahaemolyticus* strains, (ii) possible geographic preferences for predators from one area to infect vibrios from the same or other, more distant areas, (iii) the relative sizes and appearances of the plaques produced by different predatory strains on different *Vibrio* strains, (iv) predator-prey interactions based on *Vibrio* sequence types, (v) predator-prey interactions based on *Vibrio* serotypes, and (vi) predator-prey interactions as affected by the presence or absence of *Vibrio tdh* and/or *trh* hemolysin genes.

**Data availability.** *Halobacteriovorax* strains used in this study are available through a material transfer agreement (MTA) with the U.S. Department of Agriculture (USDA), while *Vibrio* strains used are available through an MTA with the U.S. Food and Drug Administration (FDA). Partial 16S rRNA sequences for the four *Halobacteriovorax* strains used in this study are provided in Table S1 in the supplemental material.

## SUPPLEMENTAL MATERIAL

Supplemental material is available online only.
**SUPPLEMENTAL FILE 1**, DOCX file, 0.01 MB.

## ACKNOWLEDGMENTS

This work was supported by USDA, ARS in-house funds under CRIS project number 8072-42000-090-000D.

*Vibrio parahaemolyticus* strains were generously provided by the U.S. Food and Drug Administration.

The use of trade names or commercial products in this publication is solely for the purpose of providing specific information and does not imply recommendation or

endorsement by the U.S. Department of Agriculture (USDA) or the U.S. Food and Drug Administration (FDA).

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
