## [Reviewer comments · Microbiology Spectrum]

Microbiology Spectrum

Predator-prey interactions between *Halobacteriovorax* and pathogenic *Vibrio parahaemolyticus* strains: geographical considerations and influence of *Vibrio* hemolysins

GARY RICHARDS, Michael Watson, Henry Williams, and Jessica Jones

Corresponding Author(s): GARY RICHARDS, United States Department of Agriculture

Review Timeline:

Submission Date:

June 12, 2023

Accepted:

June 23, 2023

Editor: Blaire Steven

Reviewer(s): The reviewers have opted to remain anonymous.

Transaction Report:

DOI: <https://doi.org/10.1128/spectrum.02353-23>

June 23, 2023

Dr. GARY P. RICHARDS
United States Department of Agriculture
Agricultural Research Service
Delaware State University
James W. W. Baker Center
Dover, DE 19901

Re: Spectrum02353-23 (Predator-prey interactions between *Halobacteriovorax* and pathogenic *Vibrio parahaemolyticus* strains: geographical considerations and influence of *Vibrio* hemolysins)

Dear Dr. GARY P. RICHARDS:

After review and transfer I am happy to accept your paper for publication in Microbiology Spectrum. Please ensure that you provide a data availability statement prior to final publication

Your manuscript has been accepted, and I am forwarding it to the ASM Journals Department for publication. You will be notified when your proofs are ready to be viewed.

Sincerely,

Blaire Steven
Editor, Microbiology Spectrum
